# Palmitoylethanolamide and White Matter Lesions: Evidence for Therapeutic Implications

**DOI:** 10.3390/biom12091191

**Published:** 2022-08-27

**Authors:** Marta Valenza, Roberta Facchinetti, Luca Steardo, Caterina Scuderi

**Affiliations:** 1Department of Physiology and Pharmacology “Vittorio Erspamer”, SAPIENZA University of Rome—P.le A. Moro, 5, 00185 Rome, Italy; 2Università Giustino Fortunato, 82100 Benevento, Italy

**Keywords:** palmitoylethanolamide, luteolin, oligodendrocyte, astrocyte, myelin, demyelinating diseases, multiple sclerosis, Alzheimer’s disease

## Abstract

Palmitoylethanolamide (PEA), the naturally occurring amide of ethanolamine and palmitic acid, is an endogenous lipid compound endowed with a plethora of pharmacological functions, including analgesic, neuroprotective, immune-modulating, and anti-inflammatory effects. Although the properties of PEA were first characterized nearly 65 years ago, the identity of the receptor mediating these actions has long remained elusive, causing a period of research stasis. In the last two decades, a renewal of interest in PEA occurred, and a series of interesting studies have demonstrated the pharmacological properties of PEA and clarified its mechanisms of action. Recent findings showed the ability of formulations containing PEA in promoting oligodendrocyte differentiation, which represents the first step for the proper formation of myelin. This evidence opens new and promising research opportunities. White matter defects have been detected in a vast and heterogeneous group of diseases, including age-related neurodegenerative disorders. Here, we summarize the history and pharmacology of PEA and discuss its therapeutic potential in restoring white matter defects.

## 1. How Nature Provides Therapeutic Molecules: The History of Palmitoylethanolamide

The discovery of palmitoylethanolamide (PEA) arises from an interesting clinical observation in the early 1940s, when some clinicians reported that adding dried chicken egg yolk to the diets of underprivileged children reduced recurrences of rheumatic fever, despite the presence of streptococcal infections [1]. Later on, anti-allergic effects of the lipid fractions purified from egg yolk [2,3], peanut oil, and soybean lecithin [4] were demonstrated in guinea pigs. The agent responsible for these properties was definitively recognized as PEA [3], which was also identified in mammalian tissues fifteen years later [5].

The following years witnessed a series of interesting clinical studies on this molecule with good results. During the late 1970s, PEA clinical use was approved in former Czechoslovakia for acute respiratory diseases. For unknown reasons, not related to its toxicity, PEA was withdrawn from clinical use after several years on the market. In fact, the molecule remained largely unnoticed by the rest of the world for more than 20 years. Indeed, the market withdrawal and the failure to identify PEA molecular targets caused a period of research stasis. In the 1990s, Nobel laureate Prof. Rita Levi-Montalcini demonstrated that PEA is an endogenously produced regulator of inflammation [6]. This evidence, together with the discovery of the endocannabinoid system, caused a renewal of interest in PEA. In particular, during the last two decades, formulations containing PEA have received increasing attention as drugs and dietary supplements to be used in chronic pain, inflammation, and certain brain diseases.

## 2. The Pharmacology of Palmitoylethanolamide

PEA exhibits a rather complex biological and pharmacological profile. Despite some structural and metabolic similarities with the endocannabinoid signaling molecules, PEA is not a classical cannabinoid. It is occasionally called a “promiscuous” molecule, as it binds several receptors and also interacts with non-receptor targets (Figure 1). This alleged lack of selectivity is, however, increasingly regarded as an advantage in certain diseases, where “multiple-target” molecules may exert more beneficial effects than classical “single-target” drugs [7].

### 2.1. Biosynthesis, Degradation, and Pharmacokinetics of PEA

PEA is an endogenous compound belonging to the family of N-acylethanolamines (NAEs) which include the endogenous cannabinoid receptor ligand anandamide (AEA, arachidonoylethanolamide) and the satiety agent oleoylethanolamide (OEA) [8,9].

Biosynthesis of PEA occurs on-demand within the lipid bilayer in two steps [10]. The first is the calcium- and cAMP-dependent transfer of palmitic acid from phosphatidylcholine to phosphatidylethanolamine to form N-acylphosphatidylethanolamine (NAPE); the second step is the cleavage of NAPE to release PEA, mediated by a NAPE-specific phospholipase D. The inactivation of PEA occurs via fatty acid amide amidohydrolase (FAAH) or PEA-preferring acid amidase (PAA) to form palmitic acid and ethanolamine.

All tissues, including the brain, have been found to contain PEA [11]. The physiological regulation of PEA levels in mammalian tissues is still under investigation. Different studies suggest that this compound accumulates after tissue injury, causing cellular stress; this supports the hypothesis that PEA is an endogenous mediator whose levels increase to exert a local reparative action [12].

Exogenous PEA administration presents some issues regarding its bioavailability. Given its lipid nature, PEA solubility in most aqueous solvents is indeed very low. It was shown that after intraperitoneal administration of radio-labeled-PEA to rats, the molecule was distributed mainly in some peripheral organs, whereas only low concentrations were detected in the brain and plasma [13]. Other groups demonstrated the ability of PEA to cross the blood–brain barrier (BBB) after oral administration, but only in very small amounts [14,15]. Of note, one clinical study reported that, in a dose-dependent manner, PEA administration leads to a two- to nine-fold increase in PEA plasma baseline concentrations [16].

To overcome the poor pharmacokinetics of PEA, different formulations containing PEA as micronized or ultra-micronized particles, or as ester derivatives (prepared by conjugating PEA with various amino acids), have been recently developed. These formulations improve PEA bioavailability in the central nervous system (CNS), without affecting its pharmacological efficacy [8].

### 2.2. The Search for the Palmitoylethanolamide Receptor

Between the 1950s and the 1980s, the mechanism(s) of action of PEA remained unidentified. This probably caused the lack of interest of the scientific community in PEA, despite its potential clinical significance. The renaissance of PEA originates from the work of Prof. Levi-Montalcini, who suggested that this endogenous compound exerts anti-inflammatory effects by serving as an autacoid local injury antagonist (ALIA) leading to a down-regulation of mast cell activation [6]. Moreover, the concurrent discovery of the endocannabinoid AEA and the cannabinoid receptors CB1 and CB2 shed new interest in PEA. Indeed, the similarity in chemical structure between AEA and PEA first suggested that these two endogenous mediators might share the same receptor. Several preclinical studies have definitely clarified that PEA does not bind the CB receptors and identified other mechanisms of action. The main molecular target of PEA seems to be the peroxisome proliferator-activated receptor-alpha (PPAR-α), through which PEA exerts its strong anti-inflammatory effects. Supporting this evidence, PEA effects vanished in mutant PPAR-α-null mice [10]. In particular, PPAR-α emerged as the key factor in mediating the ability of PEA to control neuroinflammation in different brain diseases [17,18,19,20,21,22,23,24].

Additionally, PEA can directly activate the G protein-coupled receptor 55 (GPR55) [25] and G protein-coupled receptor 119 (GPR119) [11]. Furthermore, it has been proposed that PEA indirectly potentiates the CB1 signal by inhibiting the degradation of AEA, a phenomenon known as the “entourage effect” [26,27]. Additionally, several studies have demonstrated the involvement of the transient receptor potential vanilloid 1 (TRPV1) in the actions of PEA [28,29,30,31]. The main PEA targets are schematized in Figure 1.

The pharmacological properties of PEA are numerous and widely acknowledged by the scientific community. However, among PEA properties, its therapeutic potential in pathologies characterized by myelin defects is scantly investigated. In this review, we summarize the most recent findings obtained with different formulations containing PEA in heterogeneous disease conditions, all of which are characterized by white matter defects.

## 3. Myelin Sheath Organization and Functions

The term myelin was coined by Rudolf Virchow in 1864 and derives from the Greek word “myeloid” (marrow). Myelin provides the structural basis for fast impulse propagation along neuronal axons, a process required for the proper performance of motor, sensory, and cognitive functions in the CNS [32]. The myelin sheath is an extension of the plasma membrane of mature oligodendrocytes, cells belonging to the family of glial cells [33]. Oligodendrocytes originate from oligodendrocyte precursor cells (OPCs). During development, OPCs migrate from the ventricular/subventricular zones of the CNS to the whole brain, where they proliferate [33]. Differentiation into oligodendrocytes was the first acknowledged function of OPCs [34]. However, a rather large percentage of OPCs never differentiate and live throughout adulthood, suggesting that OPCs may play additional important roles [35]. For instance, in adulthood, when a myelin lesion occurs (e.g., in cases of brain damage), local OPCs begin to proliferate and differentiate into myelinating oligodendrocytes [34,36].

The maturation of OPCs requires several stages, each characterized by distinctive morphological and functional aspects and by the expression of specific proteins (Figure 2).

Early OPCs are bipolar cells characterized mainly by the expression of the platelet-derived growth factor receptor-α (PDGFR-α). This receptor mediates the signal of the neuronal and astrocytic PDGF-α, which regulates the proliferation, migration, survival, and maturation of OPCs [37]. Once these cells keep contact with a target axon, they lose their bipolarity and start to develop filamentous myelin outgrowths, developing a complex shape. At this differentiation stage, pre-oligodendrocytes express 2′, 3′-cyclic-nucleotide 3′-phosphodiesterase (CNPase) and the cell surface markers O4 and O1. Finally, once fully mature, oligodendrocytes show a complex and ramified structure and form myelin sheaths. Myelinating oligodendrocytes can be identified by the expression of several markers including the myelin basic protein (MBP), the proteolipid protein (PLP), and the myelin-associated glycoprotein (MAG), as well as by the membrane marker galactocerebroside (GalC) and the surface marker myelin-oligodendrocyte glycoprotein (MOG) [38]. Olig2 is a transcription factor that drives oligodendrocyte-lineage cells to express myelin-related genes. It appears to be constantly expressed by oligodendrocytes regardless of the degree of differentiation, but progressively decreases at maturity [39].

The myelin structure consists of a multilayered battery of membranes organized in alternating electron-dense and electron-light layers [40]. The electron-dense layers represent the zone of adhesion between the tightly condensed cytoplasmic membranes, while the electron-light layers consist of extracellularly arranged myelin membranes [40]. The myelinated portions along axons are separated by the nodes of Ranvier, small myelin-free areas that promote rapid saltatory transmission of the nerve impulse by concentrating voltage-dependent sodium channels [41]. Myelin structure is stabilized by a multitude of adhesion mechanisms and proteins. The compaction of myelin is primarily performed by MBP, which condensates two adjacent cytoplasmic membrane surfaces into a major dense line of myelin [42,43]. The compaction begins in the very outer layers, close to the cell soma, and proceeds to the inner side in parallel with the growth of the membrane. In order to ensure the success of this process, MBP mRNA is trafficked from the nucleus to the myelin compartment where it is locally translated [44]. The integral membrane protein PLP allows the extracellular and cytoplasmic membranes to adhere to each other, enhancing the compaction of myelin [45]. The presence of the cytoplasm-rich compartments within myelin is ensured by CNPase which, by directly associating with cytoskeletal actin, antagonizes the adhesive forces exerted by polymerizing MBP [46]. Cytoplasmic channels allow the bidirectional movement of macromolecules under the myelin sheath maintaining vital such a compacted structure.

Through such channels, oligodendrocytes provide neurons with trophic and metabolic support. For example, oligodendrocytes use monocarboxylate transporters to deliver energy metabolites, like pyruvate and lactate, to neurons that use them to produce ATP [47].

## 4. The Role of the Astrocyte-Oligodendrocyte Cross-Talk in Myelination

The proliferation and differentiation of OPCs need the support of other glial cells, mainly microglia and astrocytes that release distinct patterns of secreted molecules to drive these processes. Microglia involvement in OPC differentiation, as well as in the context of myelin repair, is acknowledged, and the data in the literature highlight a dual role of these cells, providing evidence for both trophic and detrimental roles of microglia on oligodendrocytes and myelin (for a comprehensive review see [48]). Likewise, the role of astrocytes in the proliferation of OPCs and the formation and repair of myelin has long been recognized. It is believed that impairments of these star-shaped cells are implicated in the development of demyelinating diseases. This was first hypothesized by Müller in 1904, who was convinced that multiple sclerosis (MS), a demyelinating disease, was characterized by astrocytic dysfunction [49]. Research findings collected so far have confirmed this hypothesis, with various studies highlighting the key role of the cross-talk between astrocytes and oligodendrocytes in myelination, in both health and disease [50,51].

Astrocytes are the most abundant glial cell type of the CNS, found in both white and grey matter [52]. They were long considered secondary to neurons and were defined, from the Greek term glia, as “brain glue”. However, research studies over the past two decades have demonstrated that astrocytes exert a plethora of different functions to maintain CNS homeostasis at molecular, cellular, organ, and system levels of organization [53]. Astrocytes provide both physical and metabolic support to neurons [50] and modulate synaptic transmission and information processing by neural circuits [54]. Astrocytes are key components of the BBB; in this way, they regulate cerebral blood flow and the communication between the CNS and the periphery [54]. Furthermore, they are involved in several processes, such as ion and water transport, pH buffering, neuroplasticity, and synapse pruning. They also release approximately 200 molecules, including neurotrophic factors and energy substrates, thus, providing trophic and metabolic support to all cells in the CNS [55].

Astrocytes and oligodendrocytes originate from a common lineage of neural progenitor cells within the neuroectoderm [56] and, after development, communicate in physical and functional ways. Physically, astrocytes are coupled to oligodendrocytes through gap junctions at the cell body level and the paranodes, directly connecting the outer layer of the myelin sheath with an astrocytic process [57,58]. Such intercellular channels are heterologous, since oligodendrocytes express connexin (Cx) 32, Cx47, and Cx29, whereas astrocytes have Cx26, Cx30, and Cx43 [59]. A large number of reports underscore the importance of gap junctions in myelination. For instance, mutations or genetic ablation of specific connexins could lead to myelin defects, demyelinating diseases, encephalopathies, and peripheral neuropathies [60].

Besides direct physical interaction, astrocytes can communicate with oligodendrocytes by releasing a variety of soluble factors, including PDGF, brain-derived neurotrophic factor (BDNF), ciliary neurotrophic factor (CNTF), transforming growth factor (TGF)-β, and basic fibroblast growth factor (FGF2). All of these are considered promoters of OPC proliferation and survival [48], despite some contrasting evidence suggesting that PDGF and FGF2 could also act as inhibitors [61,62]. Moreover, the tissue inhibitor of metalloproteinase-1 (TIMP-1), an endogenous regulator of matrix metalloproteinases, promotes oligodendrogenesis [63]. In addition, astrocytes maintain and support myelin sheath formation by synthesizing and delivering cholesterol to oligodendrocytes [64,65].

As astrocytes are critical for maintaining brain homeostasis [50], they promptly react to any CNS insult or damage by undergoing morphofunctional changes that are context-, time-, and disease-specific [66,67,68,69]. Astrocyte reactivity could have either beneficial or detrimental consequences for myelination, and the difference seems to be related to the severity of astrogliosis [70]. Mild astrogliosis leads to the release of CNTF, FGF2, and the proinflammatory interleukin (IL)-6, all associated with OPC survival, proliferation, and maturation [71,72]. During severe astrogliosis, astrocytes secrete tumor necrosis factor (TNF)-α, which correlates with the extent of demyelination in MS and with myelin and oligodendrocytes damage in vitro [73,74]. Additionally, in an animal model of experimental autoimmune encephalomyelitis (EAE) the release of interferon (IFN)-γ was shown to suppress remyelination and delay recovery [75].

The relevance of astrocyte contribution to oligodendrocyte function is also evident in several astrocytopathies, such as Alexander disease, vanishing white matter, megalencephalic leukoencephalopathy with subcortical cysts, Aicardi–Goutières syndrome, and oculodentodigital dysplasia [76]. All these diseases are characterized by genetic mutations that cause defective astrocyte function, with major consequences for oligodendrocyte physiology as low oligodendrocyte survival, impaired myelination, and absence of myelin with or without concurrent development of astrogliosis [77,78].

Based on this evidence, astrocytes appear as the major cells orchestrating cell-to-cell communication relevant for myelination. Thus, alterations in this cross-talk could lead to myelin defects (Figure 3).

## 5. White Matter Defects in Demyelinating Diseases and the Therapeutic Potential of Palmitoylethanolamide

A vast and heterogeneous group of diseases are characterized by white matter abnormalities. Furthermore, any disorder that weakens or alters the myelin sheath, which surrounds the nerve fibers in the brain, optic nerves, and spinal cord, is referred to as a demyelinating disease. Several hereditary or acquired conditions can be distinguished, with the most common having inflammatory, infectious, toxic, and metabolic origins [36,79]. Myelination defects in humans typically result in substantial neurological symptoms, as would be predicted given its crucial involvement in the physiology of the mammalian nervous system. Indeed, when the myelin sheath is compromised, nerve impulses slow or even cease, leading to neurological issues. The most common symptoms of demyelinating disorders are vision loss, muscle weakness or stiffness, muscle spasms, and alteration in the bladder and/or bowel movements [80].

As mentioned above, a growing number of reports demonstrated the anti-inflammatory and neuroprotective effects of PEA in various models of brain diseases. This evidence opened up the possibility that PEA could be beneficial even in demyelinating diseases characterized by strong neuroinflammatory components. Among them, MS is the most common acquired demyelinating disease of the CNS [79,81]. Its etiology is not yet fully understood, but its inflammatory basis is known. The current therapeutic approach is based on the lifelong administration of immunosuppressive and immunomodulatory agents, in addition to corticosteroids to treat acute relapses [79,82]. Unfortunately, all these therapies have limited efficacy and many adverse effects. Moreover, they do not address all MS symptoms. There is, therefore, an urgent medical need for innovative therapies. In light of this, promising results were obtained in a small double-blind, randomized, placebo-controlled study in which relapsing–remitting MS patients took oral ultra-micronized PEA (um-PEA; NORMAST^®^ 600 mg/day for up to one year), or placebo, in addition to subcutaneous IFNβ1a (Rebif^®^ 132 μg/week). The um-PEA reduced pain sensation at the IFN-β1a injection site, improved results of a questionnaire assessing cognition, reduced IL-17 and TNF-α serum concentration, and increased AEA and OEA plasma levels. Overall, such chronic um-PEA supplementation did not show negative effects and the quality of life of patients seemed to improve [83].

Besides the neuroinflammatory component, oxidative stress also contributes to tissue injury in MS, and it promotes the inflammatory response [82]. For this reason, formulations containing PEA together with an antioxidant compound could be of advantage. Some in vitro studies indeed show that co-ultra-micronized palmitoylethanolamide/luteolin composite (co-ultra PEALut, 10:1 by mass) exerts beneficial effects that could be relevant for MS. For instance, pre-treatment of primary OPCs with co-ultra PEALut prevented the increase in gene expression of serum amyloid A protein (SSA) induced by challenging cells with the pro-inflammatory stimuli TNF-α [84]. This co-ultra PEALut effect is in line with the well-known anti-inflammatory effects of PEA and is relevant in both MS and Alzheimer’s disease (AD) in which an elevation in SSA has been detected [85,86].

Of note, treatment of differentiating OPCs with co-ultra PEALut increases the transcription of major myelin proteins genes, as well as genes for enzymes involved in lipid biosynthesis, in a time-dependent manner. Neither PEA nor Lut was able to produce such effects on its own [87]. Co-ultra PEALut promotes MBP and PLP expression in primary OPCs cultured either in a medium favoring cell differentiation or in a medium favoring cell proliferation, and it does so in a rapamycin-dependent manner [88]. Co-ultra PEALut time-dependently increases gene expression of the myelin-associated proteins MBP, CNPase, and the tyrosine kinase TAM receptor Tyro3 in primary OPCs cultured in a pro-differentiating medium, while it induces a reduction in the gene expression level of other two TAM receptors (Axl and Mertk) compared to control cells [89]. All these effects were blunted by concurrent treatment with rapamycin, again supporting the hypothesis that an mTOR-dependent molecular pathway is involved in the co-ultra PEALut mechanism of action [87,89,90].

Two different research groups showed in vivo the potential benefit of administering formulations of PEA. In a model of EAE, commonly used to study features of MS in mice, daily systemic co-ultra PEALut administration, initiated at the first signs of sickness, progressively improved scores for neurological assessment compared to vehicle. Lower doses of co-ultra PEALut were less efficacious. Such behavioral improvement was accompanied by a co-ultra PEALut-mediated reduction in the disease-induced gene expression level of pro-inflammatory mediators and receptors [91]. These effects were likely driven by PEA since its administration alone showed similar beneficial effects in a separate and precedent study. Indeed, PEA ameliorated histological signs of the disease in spinal cord specimens from AEA mice compared to control animals. It also reduced the AEA-increased mRNA level of proinflammatory markers in spinal cord samples [92]. Chronic administration of PEA also showed a protective effect on lesioned peripheral nerves in a murine model of neuropathic pain, as the chronic constriction injury of the sciatic nerve (CCI). Furthermore, PEA, by engaging the PPAR-α, prevented the CCI-induced thinning of the myelin sheath and reduction in axonal diameter, as well as edema and macrophage infiltration [93]. Some authors have suggested that PEA exerts its antinociceptive activity at the spinal level by inhibiting responses of dorsal horn wide dynamic range neurons [94].

Of note, in a small group of patients with chemotherapy-induced painful neuropathy, a two-month treatment with PEA produced substantial pain relief and restored myelinated-fiber function in patients [95].

Taken together, these results highlight the beneficial effects of administering PEA in various conditions which share alteration or loss of myelin, acting at various levels and through different mechanisms.

## 6. White Matter Defects in Age-Related Neurodegenerative Diseases and the Therapeutic Potential of Palmitoylethanolamide

Aging is the main risk factor for most neurodegenerative diseases. Aging occurs at different rates in diverse species, while inter-individual variations exist within a species and in the different organs and tissues of a subject. The brain is primarily composed of postmitotic cells, so it is especially sensitive to the effects of aging. The cause of the cognitive decline exhibited by human and non-human primates during normal aging has long been considered the consequence of a loss of cortical neurons. Advances in brain imaging techniques have shown that a significant number of cortical neurons are not lost [96]. Therefore, other reasons for the cognitive decline have been sought. One of the putative contributing factors could be the age-related degenerative change in the morphology of myelinated nerve fibers [97]. It has been well established that white matter volume starts to decrease gradually from 50 years of age onwards [98]. Myelin sheaths exhibit degenerative changes with age. These include the formation of splits at the major dense line with electron-dense cytoplasm inclusions and balloons [99]. These balloons appear as holes, but electron microscopy reveals that they are spherical cavities causing the myelin sheaths to swell. Moreover, increased formation of sheaths with redundant myelin and circumferential splits in thick sheaths are considered as other age-related changes. Some of these multilamellar myelin fragments represent myelin outfolding, while others are engulfed by microglia, suggesting that microglia may actively strip off damaged myelin [100]. In the aged mouse brain, microglia display an expanded lysosomal compartment and accumulate autofluorescent material which may consist of remnants of indigestible myelin. Hence, the increased burden of myelin clearance in the aging brain may also contribute to age-associated microglial dysfunction.

Structural abnormalities in the white matter have been observed also in neurodegenerative diseases, including AD and Parkinson’s disease (PD), which share symptoms, such as locomotor disorders and cognitive decline [101]. Evidence is accumulating that white matter lesions independently contribute to postural and gait disturbances typical of PD, as well as increase the risk of dementia in such patients [102,103,104,105,106]. Despite some data indicating that increased lesions of myelin are associated with worsening cognitive performance in PD, further studies are needed to examine the relationship between myelin lesions and PD-specific cognitive domains.

For a long time, AD has been considered a disease of grey matter. However, advancements in brain imaging techniques have now highlighted evident lesions to the white matter. Postmortem and in vivo imaging studies have demonstrated that AD brains show both reduced volume and microstructure alterations of the white matter, including defects in the physical organization of the myelin lipid bilayers, as reviewed in [107]. Studies have linked myelin damage and concurrent impairments in the maturation of oligodendrocytes with AD pathogenesis, appointing the cerebral deposition of beta-amyloid (Aβ) as a possible etiological factor [108,109,110]. Thus, myelin loss and the inability of oligodendrocytes to repair myelin damage have been suggested as additional central features of AD [111], besides the accumulation of toxic species of Aβ and the development of neurofibrillary tangles [112]. Biochemical analysis of total myelin fraction in AD patients revealed increased Aβ levels accompanied by a significant reduction in MBP, PLP, and CNPase, as well as alterations in myelin lipid content [113].

Despite this compelling evidence, very little is known about AD-related changes in oligodendrocytes maturation and myelin formation.

Counterintuitive effects of Aβ on oligodendrocyte differentiation have been reported. Some researchers showed that a direct treatment of oligodendrocytes with oligomeric Aβ1-42 promoted MBP upregulation in cultures of primary rat oligodendrocytes [114]. Furthermore, an accelerated differentiation of oligodendrocytes was found in different brain areas of transgenic models of AD (i.e., 3xTg-AD and APP/PS1 mice), compared with wild-type animals [115]. Very recently, we provided the very first data showing that, in a trans-well system (rat primary astrocytes and OPCs co-cultured separated by 0.4 µm pores), challenging astrocytes with Aβ1-42 increased the number of MBP^+^ oligodendrocytes [116]. However, through an in-depth morphological analysis, we revealed that this push to the maturation operated by Aβ1-42 actually induces aberrant morphological changes in oligodendrocytes. Indeed, we noted a significant reduction in the cell surface area as well as in the number of intersections, suggesting that MBP^+^ oligodendrocytes lose their proper dimension and complexity. These results agree with recent findings obtained in 3xTg-AD mice, showing that oligodendrocytes of six-month-old transgenic mice were atrophic compared with non-transgenic age-matched mice [117]. A conceivable explanation of the effects of Aβ on oligodendrocyte MBP expression could be related to the reaction of these cells to the toxic challenge [118] in an attempt to repair the damage and restore brain homeostasis [119]. To do so, oligodendrocytes must integrate a multitude of regulatory signals sent by neurons and other neuroglia cells, mainly astrocytes [120,121]. The communication between oligodendrocytes and neurons has been studied quite extensively, while their cross-talk with astrocytes has been scantly investigated in AD. As discussed in paragraph 4, growing evidence indicates that, during some pathological conditions, astrocytes modify their duties, hampering important physiological functions, including OPC differentiation and myelin sheath formation [122,123]. Thus, it is conceivable that exposure to Aβ alters the maturation of OPCs and the morphology of mature oligodendrocytes by impairing proper communication between oligodendrocytes and astrocytes. In support of such a hypothesis, we found a significant reactivity of astrocytes with the elevation of several pro-inflammatory mediators and the reduction in the production of astrocytic factors strictly implicated in OPC maturation, including FGF2 and TGF-β [116]. Our data support the hypothesis that in cases of brain damage, such as in AD, astrocytes react promptly by affecting oligodendrocytes maturation and, ultimately, myelin formation. Although the therapeutic potential of PEA in modulating different AD features has already been reported [124,125,126,127,128,129], no evidence is available so far on its effect on white matter defects.

Recent evidence, summarized in paragraph 5, shows the ability of co-ultra PEALut, but not of PEA alone, in promoting the morphological development of OPCs into mature oligodendrocytes [83,85]. In our model, we tested the effects of co-ultra PEALut. Obtained results indicate that co-ultra PEALut treatment almost completely restores the impairments caused by the exposure of cells to Aβ1-42 [116]. Moreover, we demonstrated that some of the effects of co-ultra PEAlut depend on the activation of the PPAR-α. This new and never explored ability of co-ultra PEALut in counteracting the Aβ-induced effect on oligodendrocyte homeostasis opens new and promising research opportunities.

## 7. Conclusions

Here, we discussed the potential benefits of administering PEA in white matter defects. Although still few and in some cases limited to preclinical evidence, the findings available thus far are promising. They broaden the therapeutic indications of PEA, a terrific endogenous molecule endowed with numerous pharmacological activities. Considering the safety profile of PEA, the opportunity to rapidly proceed to clinical use is reasonable.

It is the express wish of the authors to thank Doctor Francesco Della Valle, a visionary Italian entrepreneur in the pharmaceutical field who, with his enlightened competence, supported preclinical and clinical research on PEA. Dr. Della Valle has contributed significantly to scientific progress in this area of research. The world scientific community has benefited enormously from Francesco’s intuitions. We address our dearest memory to you.

## Figures and Tables

**Figure 1 biomolecules-12-01191-f001:**
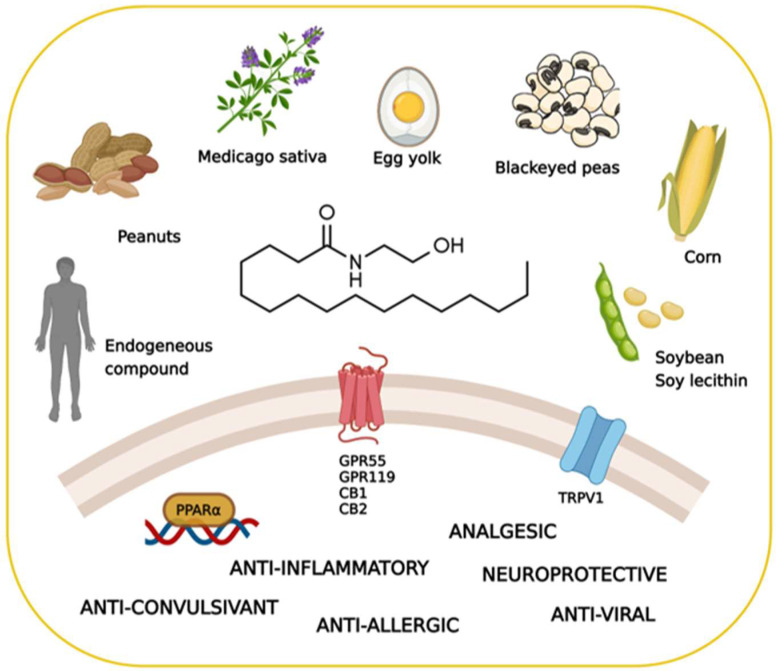
Key facts on palmitoylethanolamide (PEA), including its main sources, molecular targets, and effects. Created with BioRender.com (accessed on 22 July 2022).

**Figure 2 biomolecules-12-01191-f002:**
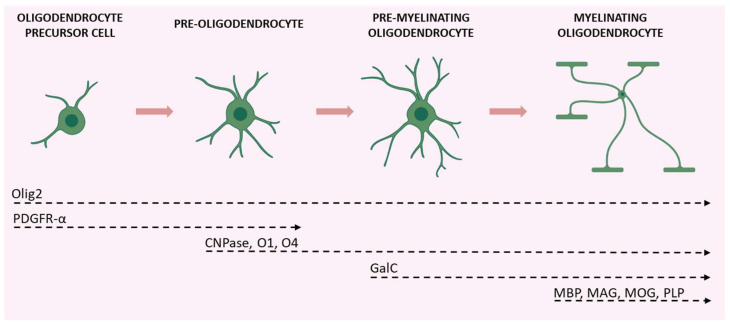
Schematic illustration of the various stages of OPC maturation and corresponding markers. Olig2 is expressed in all stages of the lineage; OPCs and pre-oligodendrocytes are characterized by PDGFR-α expression; CNPase, O1, and O4 are expressed during transition from pre-oligodendrocytes to differentiated oligodendrocytes (which also express GalC); axon-myelinating oligodendrocytes are characterized by myelin protein expression (MBP, MAG, MOG, and PLP). Created with BioRender.com (accessed on 19 August 2022).

**Figure 3 biomolecules-12-01191-f003:**
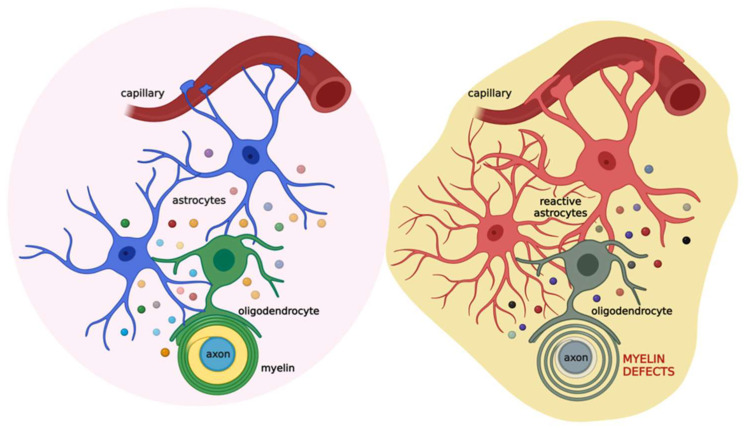
Schematic illustration of the coupling of oligodendrocytes, astrocytes, and neurons in physiological (left) and pathological (right) conditions. Astrocytes (blue) communicate with oligodendrocytes (green) through gap junctions and several released factors (colored circles), such as PDGF, BDNF, CNTF, TGF-β, FGF2, TIMP-1, ILs, TNF-α, and IFN-γ. This cross-talk allows the proper maturation of oligodendrocytes and their ability to form the myelin sheath. This, in turn, impacts neuronal activity. When brain damage occurs (see text for details), astrocytes (red) react promptly, affecting oligodendrocytes (grey) activity and ultimately the myelin sheath. Created with BioRender.com (accessed on 22 July 2022).

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
