# Peer review of "Palmitoylethanolamide and White Matter Lesions: Evidence for Therapeutic Implications"

_biomolecules, 2022, doi:10.3390/biom12091191_

Round 1

Reviewer 1 Report

This review on PEA and white matter lesions discusses previous work on PEΑ with respect to myelin as well as the therapeutic potential of PEA in restoring white matter defects. In general, it is well written and covers the aspects that may interest the readers of the journal. Some minor comments include: 

a)    Better connection for section 3. The text abruptly changes from PEA to myelin.

b)    The introduction to myelin and its structure (section 3) is very basic and I presume it is redundant knowledge

c)     I would add a scheme to illustrate the various stages of OPC maturation with the corresponding markers

d)    No discussion on microglia is included. This should be corrected as microglia involvement is very important in demyelinating pathologies. Also, there is well established cross-talk between microglia and oligodendrocytes.

e)    The following references should be added

1)Co-ultramicronized Palmitoylethanolamide/Luteolin Promotes the Maturation of Oligodendrocyte Precursor Cells

Massimo Barbierato 1Laura Facci 1Carla Marinelli 1Morena Zusso 1Carla Argentini 1Stephen D Skaper 1Pietro Giusti 1

2)Palmitoylethanolamide restores myelinated-fibre function in patients with chemotherapy-induced painful neuropathy

A Truini 1A BiasiottaG Di StefanoS La CesaC LeoneC CartoniV FedericoM T PetrucciG Cruccu

Author Response

We thank the Reviewer for his/her comments and criticisms that certainly improved the quality of our manuscript. 
Here is a point-by-point response.

a) Better connection for section 3. The text abruptly changes from PEA to myelin.

We thank the Reviewer for this comment. We added a sentence at the end of section 2 to introduce the following sections.

b) The introduction to myelin and its structure (section 3) is very basic and I presume it is redundant knowledge

We agree with the Reviewer. However, as this Special Issue covers many diverse topics and is aimed at a broad audience, we decided to add in a schematic way a brief introduction to the formation and functions of myelin. Also, accepting the c) criticism of the Reviewer, in this paragraph, we have now added the required figure on the various stages of OPC maturation with the corresponding markers.

c) I would add a scheme to illustrate the various stages of OPC maturation with the corresponding markers

According to the Reviewer's suggestion, we have included a figure on the various stages of OPC maturation with the corresponding markers in section 3.

d) No discussion on microglia is included. This should be corrected as microglia involvement is very important in demyelinating pathologies. Also, there is well established cross-talk between microglia and oligodendrocytes.

We agree with the Reviewer however, this aspect is a bit far from the target of the current review and would deserve a dedicated review. Since we agree with his/her criticism, we decided to add a couple of sentences to mention the importance of microglia on OPC differentiation as well as in the context of myelin repair during demyelinating diseases at the beginning of section 4.

e) The following references should be added

1)Co-ultramicronized Palmitoylethanolamide/Luteolin Promotes the Maturation of Oligodendrocyte Precursor Cells.

Massimo Barbierato 1, Laura Facci 1, Carla Marinelli 1, Morena Zusso 1, Carla Argentini 1, Stephen D Skaper 1, Pietro Giusti 1

2)Palmitoylethanolamide restores myelinated-fibre function in patients with chemotherapy-induced painful neuropathy.

A Truini 1, A Biasiotta, G Di Stefano, S La Cesa, C Leone, C Cartoni, V Federico, M T Petrucci, G Cruccu

We thank the Reviewer for the suggestions. However, we would like to highlight that the paper by Barbierato et al. is already present in the manuscript (in the bibliography at number 87).

Moreover, following the Reviewer's suggestion, we have added the suggested paper by Truini et al. in section 5 (reference number 95).

Reviewer 2 Report

The authors provide a review concerning the role of palmitoylethanolamide in myelination in health and diseases. This is suitable for Biomolcules Journal.

There is one comment but the authors may not describe this one if not needed. How is the Schwann cell myelination and its related diseases?

Author Response

The authors provide a review concerning the role of palmitoylethanolamide in myelination in health and diseases. This is suitable for Biomolcules Journal.

There is one comment but the authors may not describe this one if not needed. How is the Schwann cell myelination and its related diseases?

We thank the Reviewer for the comment. We are aware that Schwann's cells control myelin formation and repair in the peripheral nervous system. However, we decided to limit the review to the conditions affecting the central nervous system. For this reason, we talked only about oligodendrocytes, which are responsible for these functions in the central nervous system.